# RaDialog: Large Vision-Language Models for X-Ray Reporting and Dialog-Driven Assistance

**Chantal Pellegrini**[1,3]                                          CHANTAL.PELLEGRINI@TUM.DE
**Ege Özsoy**[1,3]                                                          EGE.OEZSOY@TUM.DE
[1] *Computer Aided Medical Procedures, TUM School of Computation, Information and Technology, Technical University of Munich, Boltzmannstr. 3, 85748 Garching, Germany.*
[3] *Munich Center for Machine Learning (MCML), Boltzmannstrasse 3, 85748 Garching, Germany.*

**Benjamin Busam**[1]                                                        B.BUSAM@TUM.DE
**Benedikt Wiestler**[2]                                                    B.WIESTLER@TUM.DE
[2] *AI for Image-Guided Diagnosis and Therapy, TUM School of Medicine and Health, Technical University of Munich, Ismaninger Str. 22, 81675 Munich, Germany.*

**Nassir Navab**[1]                                                      NASSIR.NAVAB@TUM.DE
**Matthias Keicher**[1]                                              MATTHIAS.KEICHER@TUM.DE

**Editors:** Accepted for publication at MIDL 2025

## Abstract

Conversational AI tools for generating and discussing accurate radiology reports could transform radiology by enabling collaborative, human-in-the-loop diagnostic processes, saving time and enhancing report quality. While, to this end, Large Vision-Language Models hold promise, current methods lack clinical correctness or are single-task models without conversational abilities. We propose a novel architecture and dataset to address these limitations. First, we propose a secondary image branch, explicitly focusing on structured clinical findings, improving the clinical correctness score by 13.3%. Second, we propose a catastrophic forgetting mitigation strategy and instruct dataset with variable dialog-based tasks, to enable our model to handle a multitude of different queries. RaDialog marks a foundational step toward clinical dialog systems, outperforming existing medical LVLMs by 15.0% in clinical correctness in report generation, 23.4% in interactive report correction, and is preferred by radiologists in 84.0% of cases over a comparative method. Our model and dataset are publicly available (https://github.com/ChantalMP/RaDialog, https://physionet.org/content/radialog-instruct-dataset/1.1.0/).

**Keywords:** Interactive Radiology Assistance, LVLMs, Report Generation, Chest X-Rays

## 1. Introduction

Radiology is crucial for clinical decision-making, with radiology reports serving as primary communication channel between radiologists and other clinicians, especially in the context of chest X-ray examinations, which are pivotal for identifying thoracic diseases (Johnson et al., 2019). The rising volume of imaging exams underscores the need for automated report generation, which promises to simplify reporting and support radiologists (Kaur et al., 2022). Beyond mere report generation, dialog-based assistance holds potential for a more collaborative diagnostic process between radiologists and AI-based tools.

However, while state-of-the-art methods for radiology report generation produce coherent

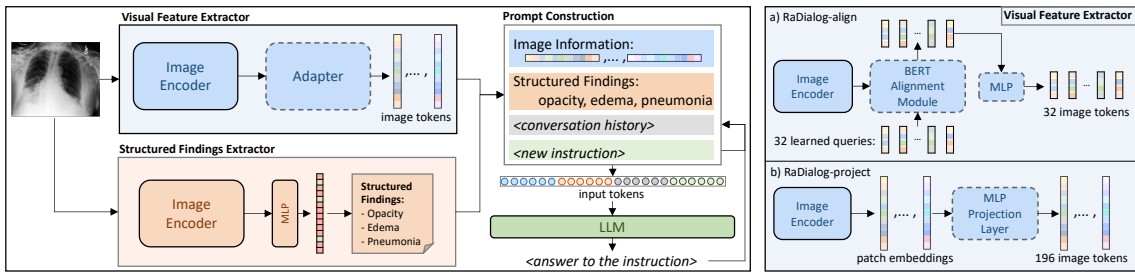

Figure 1: Pipeline overview: The Image Encoder extracts X-ray features and transforms them via adapter module a or b. The Structured Findings Extractor extracts high-level findings. Both outputs are integrated during Prompt Construction with conversation history and task-specific instructions to query the LLM. The predicted answer are added to the conversation history.

reports (Wang et al., 2022; Yang et al., 2022; Hou et al., 2023; Wang et al., 2023; Huang et al., 2023; Li et al., 2023b), they can struggle with factual correctness, and as single-task models, they are constrained to report generation as their only function.

Recent advances in large language models (LLMs) have demonstrated versatility across many tasks, including healthcare applications like medical exams and conversational diagnosis (Touvron et al., 2023; Chiang et al., 2023; Achiam et al., 2023; Singhal et al., 2023; Li et al., 2023c; Zhao et al., 2024). The development of large vision-language models (LVLMs) aims to equip these powerful LLMs with image understanding (Li et al., 2023a; Liu et al., 2024). While several previous works specifically focus on medical imaging (Tu et al., 2024; Moor et al., 2023; Li et al., 2024; Wu et al., 2023; Hyland et al., 2023; Chen et al., 2024; Thawkar et al., 2023), they are often limited to visual question-answering or single-step reporting tasks and lack robust interactive capabilities or clinical correctness. In contrast, RaDialog not only improves the accuracy of clinical report generation, but aims to enhance the radiology workflow by supporting dialog-based assistance for tasks such as report drafting, quick clarifications, collaborative insights, and reducing mental load for routine tasks. Addressing key limitations of current methods, we propose RaDialog, a collaborative radiology assistant focusing on automated report generation and auxiliary interactive downstream tasks for chest X-rays. Our key contributions include:

- A novel dual-branch architecture that, inspired by structured reporting (Pellegrini et al., 2023; Keicher et al., 2023), incorporates a secondary visual feature extraction branch to focus on structured clinical findings, leading to 13.3% improvement in clinical correctness score.
- A variable instruct training setup to enable dialog-based human-AI collaboration and combat the issue of catastrophic forgetting in LLM fine-tuning. Specifically, we design a semi-automatically labeled, image-grounded, interactive instruct dataset for X-Ray understanding, which we make publicly available.
- A context dropping augmentation, which randomly omits textual information in the conversation, requiring the model to consider the image information for all tasks.

- Demonstrated performance gains in both report generation and interactive tasks, outperforming XRayGPT (Thawkar et al., 2023) in direct comparisons, being preferred by radiologists in 84.0% of cases.

By addressing these critical aspects, RaDialog represents a significant step forward in the development of clinical dialog systems for radiology. Our work paves the way for more accurate, versatile, and user-friendly AI assistants in medical imaging.

## 2. Methodology

RaDialog leverages Large Language Models (LLMs) and visual feature extraction techniques to address the complexities of medical imaging diagnostics, particularly focusing on chest X-rays. In this section, we present our model, training, and instruct dataset.

### 2.1. Model and Training

Our architecture, visualized in Fig. 1, consists of four main components: a Visual Feature Extractor, extracting image embeddings and aligning them to the text space; a Structured Findings Extractor to capture the presence of core findings; a Prompt Construction Module; and a Large Language Model (LLM), which outputs a response given image and instruction. **Visual Feature Extractor** Given a chest X-ray image $x$, we first extract patch-wise image embeddings $x' \in \mathbb{R}^{P \times D_i}$ using a domain-specific X-ray encoder, where $P$ is the number of patches and $D_i$ is the dimensionality of each patch embedding. These patch-based features are passed to an adapter module, transforming them into $N$ embedded tokens $h \in \mathbb{R}^{N \times D_l}$, where $D_l$ is the dimension of the LLM tokens. For this adapter, we propose two variants, RaDialog-align and RaDialog-project, as depicted in Fig. 1 a) and b). RaDialog-align, inspired by the architecture of BLIP-2 (Li et al., 2023a), uses a pre-trained BERT (Devlin et al., 2018) model as alignment module, which is fine-tuned to embed the image information into $N = 32$ tokens $h \in \mathbb{R}^{N \times D_q}$, given N learned query embeddings $q \in \mathbb{R}^{N \times D_q}$ as well as the output $x'$ of the image encoder. These tokens are then projected by an MLP to retrieve $N = 32$ LLM input tokens. The alignment module is trained using three distinct objectives: an X-ray-report contrastive loss, a cross-entropy loss for image-report matching, and a language modeling loss for image-grounded report generation. This module is trained in a separate stage and remains frozen during the subsequent training of the large language model (LLM). RaDialog-project follows the image-to-text projection proposed in LLaVA (Liu et al., 2024). Here, the patch features $x' \in \mathbb{R}^{P \times D_i}$ extracted by the image encoder are directly projected to $N = 196$ language model tokens using an MLP as a projector to get the LLM input tokens $h = g(x')$ with $h \in \mathbb{R}^{N \times D_l}$. The image encoder and adapter are trained jointly with the LLM without the need for a pre-training step. **Structured Findings Extractor** In addition to direct image features, in our secondary image branch, we build a structured representation of the main clinical findings in the image $x$. This enables our model to generate a free-text report that is aligned with these explicit findings, enhancing the controllability of the output and improving the clinical efficacy of our model. The Structured Findings Extractor consists of a CLIP vision encoder, initialized with pre-trained domain-specific weights, followed by a linear classification head and is separately trained for multi-label classification. The classification output is converted into text as a comma-separated list of all positive findings. The Structured Findings Extractor

is trained using a log-weighted binary cross-entropy loss to address class imbalance.

**Prompt Construction** Given $h = \{h_j\}_{j=1}^N, h_j \in \mathbb{R}^{D_l}$, the set of image tokens obtained from the Image Encoder; $S$, the description of structured findings; $H$, the conversation history; and $I$ the instruction, the LLM input prompt is constructed. The structured findings, conversation history, and instruction are embedded by the LLM embedding layer $e(\cdot)$ into $e(S), e(H)$, and $e(I)$. The final embedded prompt $P$ is constructed as concatenation of $(h, e(S), e(H), e(I))$. This prompt effectively leverages the strengths of both the encoding and structured information about the image, while considering the conversation context.

**Language Model** Finally, the LLM processes the prompt $P$ and produces an instruction-specific response. Since the training data of generalist LLMs consists of limited medical information, we fine-tune our vision-language model on radiology reports and instructions using cross-entropy loss. This fine-tuning enhances both its medical knowledge and aligns its writing style with that of radiologists. Additionally, this fine-tuning trains the LLM to work effectively with image features and structured finding labels. For adapting the LLM, we use the parameter-efficient fine-tuning technique LoRA (Low-Rank Adaptation) (Hu et al., 2021), allowing domain adaptation with limited computational resources.

## 2.2. Instruct Dataset

Training solely on image-report pairs causes catastrophic forgetting in the LLM, reducing its ability to perform tasks beyond report generation. To address this, we design a diverse instruct dataset of ≈580k samples spanning ten tasks, each with ten prompt variations. It includes two types of tasks. Type 1 builds upon existing datasets (Johnson et al., 2019; Kayser et al., 2022; Pellegrini et al., 2023) for report generation, impression generation, CheXpert QA, Rad-ReStruct QA, natural language explanations, and view classification. Type 2 comprises replay tasks, where we, inspired by continual learning, generate pseudo-ground truth with a non-fine-tuned LLM for correction, summarization, easy language, and region QA. While imperfect, this pseudo-ground truth mitigates catastrophic forgetting by replaying general language tasks during domain-specific training. Each sample includes an image, an instruction, and the corresponding ground truth. Tasks involving reports, like summarization, integrate the report generation instruction and ground truth report in the conversation history. More details and prompt examples for all tasks are shown in appendix C.

**Context Dropping Augmentation** For tasks that can be performed with only the image, and no report, as input, including Findings QA, Region QA, and View Classification, we propose to augment the information available to the model. Context Dropping systematically imitates potential inaccuracies in the initial report by varying the availability of textual input and prevents over-reliance on the text modality, which is generally easier to interpret for the model. By training the model under different levels of textual availability, we promote feature extraction from both visual and textual information, allowing the model to develop stronger visual reasoning capabilities when answering follow-up questions even when textual context is imperfect. We specify three configurations: $c = \{full, none, partial\}$. In the *full* mode, we keep the entire report $R$, while in the *none* mode, the report and structured findings $S$ are dropped, so the model must rely on the visual encoding $h$ only. In the *partial* mode, half of the sentences in the report $R$ are randomly dropped. Let $h = \{h_ij\}_{j=1}^N$ be the set of image tokens, $S$ the structured findings, $H$ the conversation history includ-

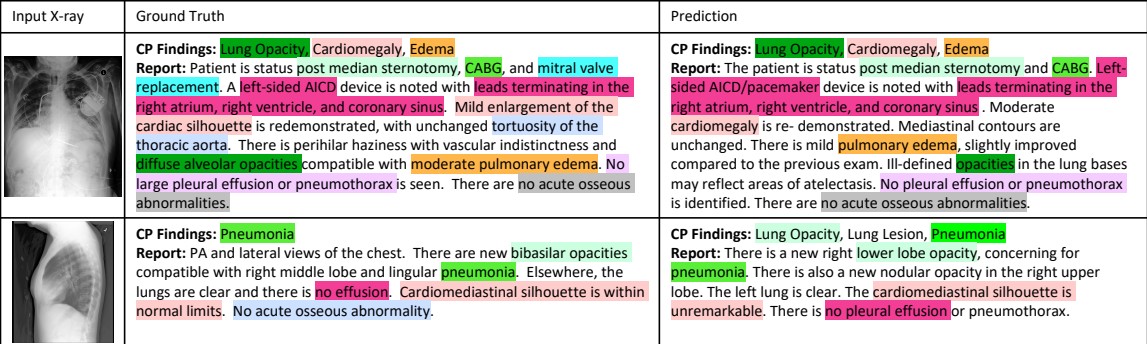

Figure 2: Qualitative report generation results of RaDialog$_{project}$ (top) and RaDialog$_{align}$ (bottom). Colors indicate matching findings in ground truth and prediction.

ing the ground truth report, and $I$ the instruction. The input prompt embedding $P$ is constructed based on the chosen configuration:

$$P = \begin{cases} \text{concat}(h, e(S), e(H), e(I)) & \text{if } c = \text{full} \\ \text{concat}(h, e(I)) & \text{if } c = \text{none} \\ \text{concat}(h, e(S), e(H'), e(I)) & \text{if } c = \text{partial} \end{cases} \tag{1}$$

where $c \sim \mathcal{U}(\{\text{full, none, partial}\})$ is randomly chosen, and $H'$ is the history with $R$ replaced by $R_{partial}$, simulating incomplete report scenarios. This augmentation technique emphasizes the model's focus on images rather than relying on the report content, enhancing its robustness and accuracy in scenarios with varying levels of report correctness.

## 3. Experimental Setup

We use the official splits of the widely used MIMIC-CXR (Johnson et al., 2019) dataset, comprising 377,110 chest X-rays and associated reports. Following prior work (Chen et al., 2020; Miura et al., 2021; Tanida et al., 2023), we predict the findings section of the reports and exclude samples with an empty findings section. For out-of-distribution evaluation, we use the test split of IU-Xray (Demner-Fushman et al., 2016). We evaluate two model types, one trained only on report generation (RaDialog$_{rep}$) and one trained on our instruct dataset (RaDialog$_{ins}$), including both report generation and interactive downstream tasks.

We evaluate clinical efficacy (CE) as macro F1 over the 14 CheXbert labels (Smit et al., 2020), embedding-based text similarity (BertScore (Zhang et al., 2019)), and standard Natural Language Generation (NLG) metrics (BLEU (Papineni et al., 2002), ROUGE (Lin, 2004), and METEOR (Lavie and Denkowski, 2009)). While conventional NLG metrics are not ideal for assessing the clinical correctness of radiology reports (Pino et al., 2021; Yu et al., 2023; Pellegrini et al., 2023), they are included for completeness. More implementation Details are provided in F.

Table 1: Comparison of RaDialog to recent medical LVLMs. FT denotes if the model was fine-tuned on the respective dataset. [a]trained in multi-view setting, but we evaluate with a single view. † re-computed results for MIMIC-CXR findings section

| | MIMIC-CXR | | | | | IU-Xray (OOD) | | | | |
|---|---|---|---|---|---|---|---|---|---|---|
| Method | FT | CE | BS | B-4 | R-L | FT | CE | BS | B-4 | R-L |
| LLaVA-Med | × | *10.7* | *0.19* | *1.1* | *15.1* | × | 5.0 | 0.20 | 1.1 | 15.8 |
| Rad-FM | ✓ | 15.4 | 0.22 | 2.4 | 15.6 | × | 5.9 | 0.20 | 2.3 | 13.8 |
| XrayGPT | ✓ | 19.3 | 0.33 | 5.4 | 22.0 | × | 9.9 | 0.39 | 5.3 | 25.7 |
| LLM-CXR | ✓ | 21.1 | - | - | - | - | - | - | - | - |
| CheXagent[a] | ✓ | 22.2 | 0.36 | 7.3 | 25.9 | ✓ | *14.1* | *0.51* | *12.7* | *34.6* |
| R2GenGPT† | ✓ | 24.7 | 0.36 | **10.1** | **27.6** | - | - | - | - | - |
| RaDialog$_{\text{align-rep}}$ | ✓ | 39.4 | **0.40** | 9.5 | 27.1 | × | 22.6 | **0.47** | 10.2 | **31.0** |
| RaDialog$_{\text{align-ins}}$ | ✓ | 38.6 | 0.39 | 9.7 | 27.0 | × | 22.9 | 0.46 | 9.7 | 30.2 |
| RaDialog$_{\text{project-rep}}$ | ✓ | **39.7** | 0.36 | 8.8 | 25.6 | × | 23.0 | 0.45 | 8.3 | 29.6 |
| RaDialog$_{\text{project-ins}}$ | ✓ | 39.2 | 0.37 | 9.4 | 26.7 | × | **23.1** | 0.45 | **11.0** | 30.4 |

## 4. Results and Discussion

**Radiology report generation**  We evaluate RaDialog on radiology report generation and compare it to recent LVLM-based methods and foundation models on MIMIC-CXR and IU-Xray in Tab. 1. RaDialog outperforms all other methods significantly in clinical efficacy (CE) and BertScore (BS). The out-of-distribution (OOD) evaluation on IU-Xray, on which neither RaDialog nor most of the compared methods were trained, further showcases RaDialogs' benefit over prior LVLMs in X-ray report generation. Fig. 2 shows qualitative report generation results on a frontal and lateral chest X-ray. The color coding was verified by three board-certified radiologists. It can be observed that both our model variants capture a majority of the findings. RaDialog achieves an inference speed of 112 tokens/second for RaDialog$_{\text{align}}$ and 223 tokens/second for RaDialog$_{\text{project}}$, resulting in an average generation time for an entire report of 1.2 or 0.6 seconds respectively. This speed is more than sufficient for integration into a radiologist's workflow, where report generation typically takes up to several minutes. The appendix includes additional comparisons to older, non-LVLM-based methods (A) and closed-source models using the indication as input (E.1), reinforcing RaDialog's strong performance.

**Ablation of Architectural Components**  Table 2 presents an ablation study evaluating the impact of fine-tuning the LLM and incorporating structured and visual image information. Comparing the first two rows, fine-tuning the LLM significantly improves performance, highlighting the importance of domain-specific adaptation. Furthermore, the results from rows two to four demonstrate that both structured and visual inputs contribute to performance gains, with the best results achieved when both modalities are combined.

**Interactive Downstream Tasks**  Apart from report generation, we further evaluate our model on different interactive downstream tasks.
**Impression Generation:** Given the ground truth findings section, we generate the impression. Tab. 3 shows that our instruct training is crucial for this, outperforming the

Table 2: Ablations of architectural components: compares using a non-fine-tuned LLM (NF) and the effect of visual (V) and structured (S) input. RaDialog-SFE refers to the classification metrics of the Structured Finding Extractor (SFE) of RaDialog.

| Method | V | S | CE | BS | B-1 | B-4 | MTR | R-L |
|---|---|---|---|---|---|---|---|---|
| RaDialog-align-NF | × | ✓ | 35.8 | 0.20 | 5.5 | 0.4 | 4.7 | 11.7 |
| RaDialog-align-report | × | ✓ | 37.3 | 0.39 | 32.6 | 8.2 | 12.8 | 25.9 |
| RaDialog-align-report | ✓ | × | 26.1 | 0.39 | 31.3 | 9.0 | 13.0 | **27.1** |
| RaDialog-align-report | ✓ | ✓ | **39.4** | **0.40** | **34.6** | **9.5** | **14.0** | **27.1** |
| RaDialog-SFE | - | - | 31.7 | - | - | - | - | - |

Table 3: Results on Impression Generation and View Classification.

| Method | Impression Generation | | | | View Classification | |
|---|---|---|---|---|---|---|
| | B-1 | B-4 | MTR | R-L | Accuracy | F1 |
| CheXagent | - | - | - | 40.3 | **97.5** | - |
| RaDialog$_{project-rep}$ | 15.7 | 3.6 | 13.4 | 18.1 | 8.0 | 7.3 |
| RaDialog$_{project-ins}$ | **40.0** | **19.5** | **19.9** | **45.8** | 97.1 | **95.9** |

report-only model and CheXagent (Chen et al., 2024) significantly.

**Report Correction:** For a quantitative evaluation, we generate correction prompts for the entire MIMIC-CXR test set, asking to correct all incorrect pathologies found by the CheXbert labeler (Smit et al., 2020) in the initial predictions. Tab. 4 shows the improvement through correction, indicating that report correction leads to an improvement of the report of around 33%, which is significantly higher than for our report-only models (10-25%) and XRayGPT (10%), the only other report generation method allowing interactive prompting.

**Finding Prediction:** We ask the model to predict the main CheXpert findings for an image in either "binary" or "complete" mode. For the binary task, we ask for a single finding and check if the answer contains "yes" or "no". For the complete prediction, we ask for a list of all findings and check for all occurrences of the 14 CheXpert labels. The results in Tab. 4 indicate that the report-only models fall short on these tasks. In contrast, both our instruct models and XrayGPT show better performance, where our instruct models exhibit significantly superior results, emphasizing its high clinical correctness.

Table 4: Findings QA and report correction results. "Initial", "Corrected", and "Δ" represent CE scores before and after correction, and the resulting improvement.

| Method | Findings QA | | | | | | Report Correction | | |
|---|---|---|---|---|---|---|---|---|---|
| | Binary Mode | | | Complete Mode | | | Initial | Corrected | Δ |
| | F1 | Prec | Rec | F1 | Prec | Rec | | | |
| XrayGPT | 20.6 | 15.4 | 42.5 | - | - | - | 19.3 | 29.3 | 10.0 |
| RaDialog$_{align-rep}$ | 1.8 | 17.3 | 7.5 | 9.8 | 16.0 | 12.5 | 39.4 | 49.9 | 10.5 |
| RaDialog$_{align-ins}$ | **39.7** | 37.5 | **43.5** | 40.3 | 39.9 | 42.0 | 38.6 | 71.7 | 33.1 |
| RaDialog$_{project-rep}$ | 29.3 | 33.0 | 30.0 | 16.2 | 28.0 | 28.1 | **39.7** | 64.7 | 25.0 |
| RaDialog$_{project-ins}$ | 36.1 | **38.0** | 38.7 | 40.1 | 39.6 | **42.0** | 39.2 | **72.6** | **33.4** |

**Rad-ReStruct QA:** We evaluate our method on the Rad-ReStruct (Pellegrini et al., 2023) dataset, a visual question-answering benchmark designed to populate structured radiology reports by answering hierarchical questions. Following the dataset's proposed procedure, we iteratively construct a conversation by asking the model about findings and their attributes, incorporating prior questions, answers, and dataset-provided answer options into the prompt. Our instruct model achieves an F1 score of 29.5, a precision of 61.8, and a recall of 41.6, surpassing the recall of the specialized hi-VQA model (F1: 32.0, precision: 64.6, recall: 33.3) while maintaining competitive precision and F1 scores. We see a clear improvement compared to our report-only model (F1: 28.7, precision: 98.1, recall: 28.8). While the report-only model has a similar F1 score, this is mainly caused by the high precision, which is caused by the high majority of negative answers in the dataset, while the model almost always predicts a negative answer. The good results of RaDialog$_{ins}$ opena path towards using specialized LVLMs for structured reporting of detailed findings.

**View Classification:** We ask the model given only an image, from which view this image was taken. We follow the experiment setup in CheXagent (Chen et al., 2024), and reach an almost perfect score in the task (see Tab. 3, again showing RaDialog's ability to ground conversation answers on the image.

**Qualitative Conversation Results** Fig. 3 shows conversation examples covering multiple tasks. Some of the tasks, such as correction and easy language, were part of our instruct dataset, while others, such as translation to another language and knowledge questions, were not seen during training. This shows how our training adapted the model to radiology-specific tasks while maintaining general capabilities of the used LLM.

Additionally, a board-certified radiologist compared both the report generation and conversational performance of RaDialog$_{align-ins}$ to XrayGPT using 50 randomly selected X-rays from the MIMIC-CXR test set. For each image, we asked both models to write a report and perform one of these conversational follow-up tasks: report correction, easy language, binary findings QA, summarization, translation to German, knowledge QA (e.g. "What is an edema?") and recommendation of follow-up diagnostics or treatment. For each image, reports were generated by both RaDialog$_{align-ins}$ and XrayGPT. The board-certified radiologist was presented with the original X-ray alongside the two generated reports, displayed in a randomized order to prevent bias. Without knowing the source of each report, the radiologist completed a structured three-question questionnaire per case, assessing their preference with regards to (1) report generation quality in terms of accuracy, completeness, and coherence, (2) conversational performance based on contextual relevance and correctness, and (3) overall. The radiologist consistently favored RaDialog across all three evaluation criteria with a preference rate of 71% in Conversational Performance and 84% in Report Generation as well as Overall. This highlights the advantages of our approach, including the integration of structured findings, fine-tuning of the LLM with image data, and employing instruct training with replay examples, while X-rayGPT fine-tunes on text-only conversations in advance, keeping image encoder and LLM are frozen during alignment.

**Comparison of RaDialog$_{align}$ and RaDialog$_{project}$** Even though the architecture and training procedure RaDialog$_{project}$ is more straightforward, both adapter types have a very similar performance in report generation and the evaluated downstream tasks. However, considering qualitative conversational performance of both models it can be observed the

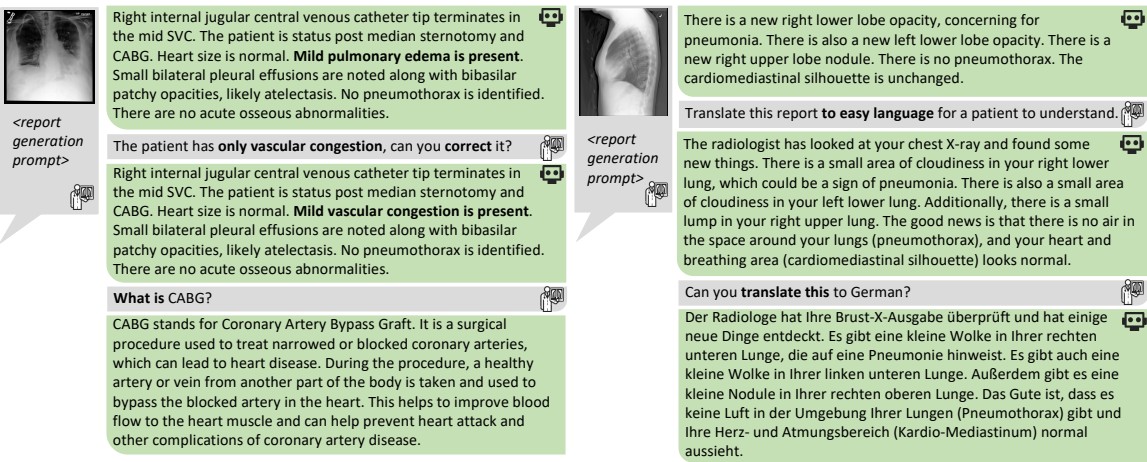

Figure 3: Qualitative conversation examples with RaDialogproject-ins (left) and RaDialogalign-ins (right), showing examples of correction, knowledge QA (zero-shot), easy language, and translation (zero-shot).

*project* variant is more robust to zero-shot tasks, we did not explicitly train on, such as knowledge questions or treatment suggestions. An example is provided in appendix D.

**Limitations** We performed an additional failure case analysis on 50 RaDialog predictions on the MIMIC-CXR test set. We observe that in general, Radialog performs well in identifying major radiological findings and the presence of support devices, showcasing its ability to detect significant abnormalities and medical hardware. The most frequent errors are mischaracterizing less frequent findings, misjudging the severity of conditions, and occasionally introducing minor incorrect details such as the exact position of findings or devices. While not flawless, Radialog can provide valuable utility for handling simpler cases, providing initial drafts and enabling collaborative radiologist-AI reporting.

## 5. Conclusion

In this work, we introduced RaDialog, a novel large vision-language model for the generation of radiology reports and auxiliary interactive assistance. Besides accurate report generation abilities, our model can also engage in a dialog, answer follow-up questions, and incorporate feedback, enabling intuitive quality control through experts in the loop. By incorporating intermediate structured radiology findings in a secondary image branch, our method reaches state-of-the-art results in creating clinically accurate reports. Secondly, through our instruct training setup on our publicly available instruct dataset dialog-based assistance is enabled, by avoiding catastrophic forgetting and teaching domain-specific conversational tasks. Lastly, by augmenting our dataset with context dropping, we enforce attention on the input image throughout the entire conversation. We believe RaDialog represents a significant leap forward from static automated report generation to a more dynamic, collaborative tool that mirrors the interactive nature of clinical practice and encourages the community to explore more collaborative medical image understanding approaches.

## Acknowledgments

This work was supported in part by the Federal Ministry of Education and Research of Germany (BMBF) under project DIVA (13GW0469C), the Bavarian Ministry of Economic Affairs, Regional Development and Energy (StMWi) under project ThoraXAI (DIK-2302-0002), and the German Research Foundation (DFG, grant 469106425 - NA 620/51-1).

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

## Appendix A. Additional report generation results

We evaluate RaDialog on radiology report generation and compare other traditional single-task report generation methods evaluated on the findings sections of the official MIMIC-CXR test set in Tab. 5. We do not include methods that use incomparable ground truth

Table 5: Performance comparison of RaDialog to existing methods on MIMIC-CXR (Johnson et al., 2019) with respect to CE and NLG metrics.

| Method | CE | BS | B-4 | MTR | R-L |
|---|---|---|---|---|---|
| R2Gen (Chen et al., 2020) | 27.6 | $0.27^a$ | 10.3 | 14.2 | 27.7 |
| MDT+WCL (Yan et al., 2021) | 29.4 | $0.28^a$ | 10.7 | 14.4 | 27.4 |
| $M^2$ Tr. (Nooralahzadeh et al., 2021) | 30.8 | $0.39^a$ | 10.7 | 14.5 | 27.2 |
| ITA (Wang et al., 2022) | 30.8 | - | 12.1 | 14.7 | 28.4 |
| METransformer (Wang et al., 2023) | 31.1 | - | 12.4 | 15.2 | 29.1 |
| Kiut (Huang et al., 2023) | 32.1 | - | 11.3 | 16.0 | 28.5 |
| M2KT (Yang et al., 2023) | 35.2 | - | 11.1 | - | 27.4 |
| COMG (Gu et al., 2024a) | 34.5 | - | 10.4 | 13.7 | 27.9 |
| HKRG (Wang et al., 2025) | 33.9 | - | **14.3** | **16.7** | **31.0** |
| ORID (Gu et al., 2024b) | 35.2 | - | 11.6 | 15.0 | 28.4 |
| MPO (Xiao et al., 2024) | 35.3 | - | 13.9 | 16.2 | 30.9 |
| RaDialog-align-report | 39.4 | **0.40** | 9.5 | 14.0 | 26.7 |
| RaDialog-align-instruct | 38.6 | 0.39 | 9.7 | 13.6 | 27.0 |
| RaDialog-project-report | **39.7** | 0.36 | 8.8 | 14.4 | 25.6 |
| RaDialog-project-instruct | 39.2 | 0.37 | 9.4 | 14.2 | 26.7 |

[a] values reported from (Jeong et al., 2023)

in terms of used test split (Tanida et al., 2023), CE score definition (Hou et al., 2023) or reports (Yang et al., 2022). RaDialog outperforms all prior works in the clinical efficacy metric, demonstrating our model's ability to infer a correct clinical diagnosis. We also outperform previous methods in the BertScore, indicating that RaDialog often predicts correct content even if the formulation differs. We hypothesize that while an LLM understands context and semantics more deeply, a smaller model trained only on a specific dataset may mirror the dataset's exact wording more closely, resulting in higher NLG scores without necessarily improving clinical correctness.

## Appendix B. Details on LVLM Configurations

Table 6 provides a detailed comparison of the configurations of all evaluated LVLMs. RaDialog does not rely on a significantly larger or more powerful image encoder, nor does it process substantially more image tokens or utilize a larger LLM compared to other models. Instead, its key differentiating factors lie in the end-to-end fine-tuning of both the image and encoder and the LLM, the proposed dual-branch architecture and the task-specific training via the RaDialog-Instruct dataset, which are designed to enhance medical report generation and diagnostic conversational abilities.

Table 6: Comparison of configurations of RaDialog and other medical LVLMs. Abbreviations: img enc. = image encoder, tokens/img = image tokens per sample, LLM = large language model, ds = datasets.

| Method | Img enc. | LLM size | Tokens per img | Datasets | end-to-end training LLM/img enc. |
|---|---|---|---|---|---|
| LLaVA-Med | CLIP-ViT | 7B | 576 | PMC-15M | ✓ / × |
| Rad-FM | 3D ViT | 13B | 32 | mix of 18 datasets | ✓ / ✓ |
| XrayGPT | MedClip | 7B | 512 | MIMIC-CXR, IU-Xray | × / × |
| LLM-CXR | VQ-GAN | 3B | 256 | MIMIC-CXR | ✓ / × |
| CheXagent | EVA-CLIP-g | 7B | 128 | mix of 28 datasets | ✓ / ✓ |
| R2GenGPT | Swin-T | 7B | 49 | MIMIC-CXR | × / ✓ |
| RaDialog$_{align}$ | BioVil-T | 7B | 32 | MIMIC-CXR, RaDialog-Instruct | ✓ / ✓ |
| RaDialog$_{project}$ | BioVil-T | 7B | 196 | MIMIC-CXR, RaDialog-Instruct | ✓ / ✓ |

## Appendix C. Instruct Dataset Details

### C.1. Task Descriptions

**Report Generation:** Produce the findings section of a radiology report given an X-ray. We use the MIMIC-CXR dataset (Johnson et al., 2019) as ground truth.

**Impression Generation:** Given the findings section of a radiology report, write the corresponding impression section. The ground truth impression sections are extracted from the MIMIC-CXR dataset (Johnson et al., 2019).

**Findings QA:** Answer a question about the CheXpert labels by either listing all findings (complete) in the image or providing a yes/no answer about a specific finding (binary). We employ MIMIC-CXR CheXbert (Smit et al., 2020) labels for supervision.

**Rad-ReStruct QA:** Answer detailed questions about the existence, location, and appearance of various chest X-ray findings in order to construct a structured report. As ground truth questions and answers, we use the samples from the Rad-ReStruct (Pellegrini et al., 2023) dataset. As this dataset is highly imbalanced, we restrict our training data to include the same number of positive and negative samples for each question type.

**Region QA:** Answer a question about a specific region, such as the lungs, which can be binary or open-ended. The supervision signal is LLM-generated.

**Easy Language:** Reformulate the produced report into a simpler and more understandable language. The supervision signal is LLM-generated.

**Summarization:** Summarize the report as bullet points or a short text. The supervision signal is LLM-generated.

**Correction:** Correct an error in the produced report. The training samples are generated by detecting wrong or missing CheXpert labels in predicted reports and asking the non-fine-tuned LLM for a corrected version.

**Natural Language Explanation:** Explain which part of the report indicates a specific pathology. We use the Mimic-NLE dataset (Kayser et al., 2022) as ground truth.

**View Classification:** Specify from which view (AP, PA, LL or lateral) the image was taken. The ground truth is collected from the metadata in the MIMIC-CXR dataset (Johnson et al., 2019).

### C.2. Instruction Prompts

We provide the exact prompts we use for report generation and three example prompts for the other tasks in the instruct dataset. The entire list of ten prompts per task will be included in our github repository.

**Report Generation:**
Image information: . Predicted Findings: <FINDINGS>. You are to act as a radiologist and write the finding section of a chest x-ray radiology report for this X-ray image and the given predicted findings. Write in the style of a radiologist, write one fluent text without enumeration, be concise and don't provide explanations or reasons.

**Impression Generation:**
• What is the impression of this radiology report?
• Summarize the radiology report findings into an impression section.
• Can you formulate the impression section based on the radiology report's findings?

**Complete CheXpert QA:**
• List all the finding in this report.
• Enumerate the observations from the report.
• What findings can be identified from this report?

**Binary CheXpert QA:**
• Is there evidence of <PATHOLOGY>in the report?
• Is there any <PATHOLOGY>?
• Does the patient have <PATHOLOGY>?

**Rad-ReStruct QA:**
• [...] Answer with one of the following options: yes, no. Question: Is there pulmonary atelectasis in the lung?
• [...] From the given list, name all correct options: left lower lobe, left upper lobe, middle lobe, right lower lobe, right upper lobe. Question: In which part of the body?
• [...] From the given list, name all correct options: focal, no selection, patchy, round, scattered, small, streaky. Question: What are the attributes?

**Region QA:**
• Is the patient's heart healthy?
• Does the patient have any abnormalities in the osseous structures?
• Are there any abnormalities in the lungs?

**Easy Language:**

- Explain this report in very easy terms, such that a child would understand.
- Given this chest xray report, formulate it in easy language.
- Reformulate this report in simple and understandable language.

**Summarization:**

- Summarize this report with bullet points.
- Provide a short summary of the most important points in this chest x-ray report.
- Please summarize this report in one sentence.

**Correction:**

- The patient also has <PATHOLOGIES>, correct the report.
- There is no <PATHOLOGIES>, please adapt the report accordingly.
- I disagree with the generated report, I think the patient has <PATHOLOGIES_1>, but does not have <PATHOLOGIES_2>. Please adapt the report.

**Natural Language Explanations:**

- What are the indicators for <PATHOLOGY>in the report?
- Why do you think the patient has <PATHOLOGY>?
- Which symptoms led to the diagnosis of <PATHOLOGY>?

**View Classification:**

- From what view was this image taken?
- Can you identify the view of this x-ray image?
- Can you specify the angle from which this x-ray scan was taken?

| Input X-ray | Ground Truth Report | RaDialog | XrayGPT | GPT-4o | Gemini-2.0 |
|---|---|---|---|---|---|
| | **Report:** Cardiac silhouette is enlarged but unchanged. There is left-sided XXXX central line with a XXXX lumen. Poly vasculature is within normal limits. Mediastinum is normal. Bibasilar opacity, left greater than right is appreciated. Left pleural effusion with adjacent atelectasis, right effusion is also present. No pneumothorax. | **A:** AP upright and lateral views of the chest provided. Left chest wall Port-A-Cath is again seen with its tip extending into the region of the right atrium. Midline sternotomy wires and mediastinal clips are again noted. There is a small left pleural effusion with left basal atelectasis. The heart is mildly enlarged. The mediastinal contour is stable. No pneumothorax. Bony structures are intact. | **A:** Based on the chest x-ray, there is an elevation of the left hemidiaphragm, with an associated opacity of the left lower lobe that is concerning for left lower lobe pneumonia. There is no pleural effusion, pneumothorax or pulmonary edema. The left-sided dual-lumen central venous catheter has been positioned into the right atrium, with the catheter tip visible within the left atrium. The heart size is normal, and the mediastinal silhouette is within normal limits. | **A:** The frontal chest radiograph demonstrates bilateral patchy opacities predominantly in the lower lung fields with obscuration of the diaphragmatic contours. Diffuse interstitial and alveolar infiltrates are present. The cardiac silhouette is mildly enlarged. Multiple lines and tubes are visualized, including endotracheal and nasogastric tubes, with their tips in appropriate positions. There is no evidence of pneumothorax or large pleural effusion. The osseous structures appear intact without acute abnormalities. | **A:** The chest x-ray demonstrates bilateral airspace opacities concerning for pulmonary edema, obscuring the cardiac silhouette. There are numerous overlying lines and tubes. No pneumothorax identified. |

Figure 4: Qualitative report generation comparison of RaDialog with XrayGPT and GPT4-Vision.

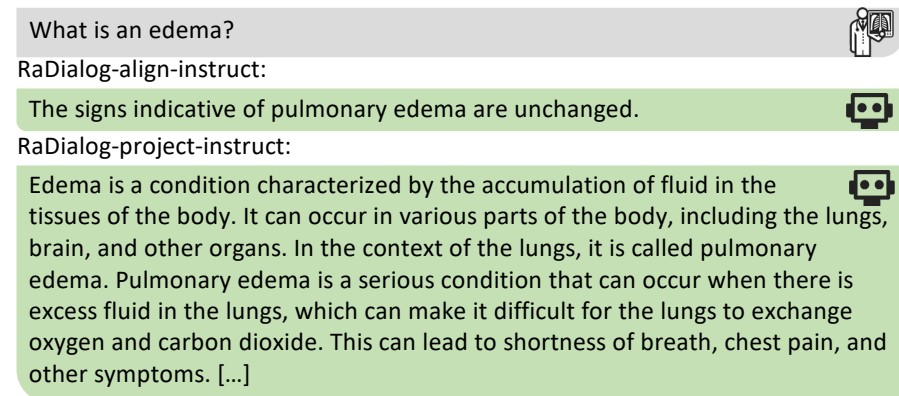

Figure 5: Differences in conversation behavior of RaDialog-align-instruct and RaDialog-project-instruct in zero-shot conversational tasks.

Table 7: Comparison to MedPaLM (Tu et al., 2024) and MAIRA-1 (Hyland et al., 2023), both closed source models using indication (Ind.) as input, compared to training on publicly available data, allowing also to publish the model.

| Method | Public | Ind. | CE | B-1 | B-4 | R-L |
|---|---|---|---|---|---|---|
| MAIRA-1 (Hyland et al., 2023) | × | ✓ | 38.6 | **39.2** | 14.2 | 28.9 |
| MedPaLM-12b (Tu et al., 2024) | × | ✓ | 37.3 | 30.9 | 10.4 | 26.2 |
| MedPaLM-84b (Tu et al., 2024) | × | ✓ | **39.8** | 32.2 | 11.3 | 27.3 |
| RaDialog-align-report | ✓ | × | 39.4 | 34.6 | 9.5 | 27.1 |
| RaDialog-align-report | ✓ | ✓ | 39.2 | **39.2** | **14.8** | **31.6** |

## Appendix D. Additional Qualitative Results

In Fig. 4, we provide a qualitative comparison of RaDialog's performance to XrayGPT (Thawkar et al., 2023), GPT-4o, and Gemini-2.0 on an out-of-domain image from the IU-Xray dataset (Demner-Fushman et al., 2016), all prompted for report generation. The prompt details to write the findings section of a radiology report, in a concise style like a radiologist. All models are aware of the correct style for report writing, but XrayGPT gets fewer findings correct and hallucinates more, while both GPT-4o and Gemini-2.0 miss most of the findings, whereas RaDialog identifies almost all of them correctly. This underlines the importance of developing domain-specific models targeted at clinical correctness. Further, Fig. 5 shows an example of the benefits of RaDialog$_{\text{project-ins}}$ in interactive abilities in zero-shot knowledge question answering compared to RaDialog$_{\text{align-ins}}$.

Table 8: Effect of different LLM sizes on report generation performance of RaDialog-align. Sec. denotes the average number of seconds to generate one report

| LLM size | Sec. | CE | BS | B-1 | B-4 | MTR | R-L |
|---|---|---|---|---|---|---|---|
| Vicuna-7b | **1.2** | **39.4** | **0.40** | 34.6 | **9.5** | 14.0 | **27.1** |
| Vicuna-13b | 1.9 | **39.4** | 0.39 | 34.8 | **9.5** | 14.0 | **27.1** |
| Vicuna-33b | 7.9 | 39.0 | **0.40** | **35.0** | **9.5** | **14.1** | 27.0 |

## Appendix E. Ablation studies

### E.1. Indication-based Methods

We compare our model to MedPaLM (Tu et al., 2024) and MAIRA-1 (Hyland et al., 2023) in Tab. 7. We separated this comparison because, unlike other state-of-the-art methods, these two use the indication section of the report as input. For comparison, we evaluate RaDialog with this additional input information and show that using the indication section leads to a significant jump in performance in the NLG metrics. Even though MedPaLM and MAIRA-1 rely on image and text encoders pre-trained with large-scale private data, we outperform MedPaLM-12b and MAIRA-1 (7b parameters) in all metrics and the 84b variant in the text-based metrics while having comparable clinical efficacy.

### E.2. Impact of Model Size

Comparing different sizes of the LLM (Tab. 8), we observe that just scaling up the LLM size does not lead to a relevant performance increase, while leading to a slower inference time. Therefore, we opt to use the seven billion parameter version for our experiments, leading to faster training and inference speeds.

## Appendix F. Implementation Details

We initialize our LLM with vicuna-7b (Chiang et al., 2023) and fine-tune it using LoRA (Hu et al., 2021) with a learning rate (LR) of $3 \times 10^{-4}$ for one or four epochs (RaDialog-align-instruct/report) on a single Nvidia A-40 GPU with 48GB memory. For RaDialog-project, we use an LR of $2 \times 10^{-5}$ and train up to five epochs with early stopping. BioVil-T (Bannur et al., 2023) is fine-tuned for multi-label classification of CheXbert findings (Smit et al., 2020) using log-weighted cross-entropy loss (six epochs, LR $5 \times 10^{-5}$) and employed for visual feature extraction. RaDialog-align uses BERT (Devlin et al., 2018) to align text and image features, trained with cosine annealing LR ($1 \times 10^{-5}$ to $1 \times 10^{-4}$) and linear warmup over four epochs.

