# OpenReview forum: "RaDialog: Large Vision-Language Models for X-Ray Reporting and Dialog-Driven Assistance"
_MIDL.io/2025/Conference — MIDL 2025 Poster_

### Official Review · Reviewer_PiCP · 2025-02-08

**Confidence:** 3
**Preliminary Rating:** 3
**Recommendation:** Poster
**Final Rating:** 4

**Summary:**

This paper introduces RaDialog, a LVLM designed for X-ray reporting and dialog-based assistance. The authors fintuned LLaVA & BLIP style models on a curated instruct dataset. Some added method include: 2nd image encoder to obtain structured report, context dropping augmentation to focus on image information. The model outperforms SoTA by a significant margin on a variety of tasks and settings.

**Strengths:**

- Pretty thorough experiments that evaluate the model on a variety of tasks and settings (e.g., ood, radiology preferences).
- The model proposed is SoTA by a significant margin, though unclear if the baselines are indeed SoTA (more in Weaknesses)
- The instruct dataset would be valuable to this community.

**Weaknesses:**

- I wonder how the proposed method compare with SoTA LVLMs like Gemini-2, GPT-4o, etc. It would be super informative because these models are general and quite good. I think a baseline with SoTA LVLMs where authors put enough effort in designing the prompt would be helpful to understand this paper's contribution.
- Techniques proposed in this paper, e.g., image branch to predict structured findings, context dropping, might be a bit too specific and therefore challenging to adopt widely beyond this specific paper.
- This paper compares method without controlling for a variety of factors, e.g., LM, LM size, the image encoder, how many tokens are assigned to represent an image, the instruction dataset used for finetuning. These factors significantly affects the performance of the model, and it would be helpful to at least tabulate this information for baselines and proposed model and discuss briefly. If the authors can clearly tell the reader how each of these factors contribute to the performance gain, it would be very helpful.

**Detailed Comments:**

- The second image encoder is related to literature on mixture of image encoders in LVLMs that the authors might find interesting.

**Justification Of The Final Rating:**

The authors did not provide comparison to state-of-the-art models, e.g., gemini-2, gpt-4o, that would be very important to know where the proposed method sits currently. In addition, the comparison is purely qualitative. The authors just cannot provide conclusion about a ranking of different methods based on a single sample.

I'm not convinced that structured findings and context dropping is a general method beyond the way it's used in the paper.

The request to provide more information on LM size, number of tokens, finetuning data size are not sufficiently addressed. I'm not asking for ablation on LM size for the proposed model, but more information on what these factors are for both the baselines and the proposed approach. The performance comparisons are misleading without these information.

Therefore, I'll keep my rating Borderline.


The author has since addressed most issues. I thank a lot for the reviewer to be responsive and proactive. And I'll gladly raise the rating to Weak Accept. Thanks!

**Justification Of The Preliminary Rating:**

RaDialog shows strong performance in X-ray reporting but lacks comparisons to top LVLMs like GPT-4o or Gemini-2, making its improvements hard to attribute. Uncontrolled factors (model size, image encoder, dataset) further obscure the source of gains. While valuable, its proposed techniques (2nd image encoder, context dropping) seem highly specialized, limiting broader impact. Clarifying these points would strengthen the paper.

**Questions To Address In The Rebuttal:**

If the author could meaningfully address the Weaknesses, I'm willing to raise the rating.

**Special Issue:**

No

---

> ### Author Response · Authors · 2025-03-06
> **Addressing the Rebuttal Questions**
>
> We thank the reviewer for recognizing the thoroughness of our experiments, the state-of-the-art performance of RaDialog, and the value of our instruct dataset to the community. We address your insightful questions below to further strengthen the paper.
>
> ### [1] “comparison to SoTA LVLMs like Gemini-2, GPT-4o”
>
> > Initially, we only compared to GPT-4 in Fig 4 in Appendix C. But we agree with the reviewer and extended the comparison to now include the more recent models, GPT-4o and Gemini-2.0 in the updated Fig 4 (Appendix C). While we see an improvement compared to GPT-4, they still fall short compared to RaDialog. Specifically, both models miss most of the findings, whereas RaDialog identifies almost all of them correctly. This underscores the value of our domain-specific fine-tuning and dual-branch architecture tailored to chest X-ray interpretation. Furthermore, unlike cloud-based models such as ChatGPT, RaDialog could be used locally within the hospital. This allows sensitive patient information to remain secure and compliant with hospital data policies.
>
> ### [2] “structured findings, context dropping, might be a bit too specific”
>
> > We believe the techniques introduced in this paper could also be valuable in different contexts. For example, while the exact contents of the structured findings would be different for a different task, application or modality, the concept of providing high-level guidance to a language model to reduce hallucinations can be helpful. Similarly, context dropping is a general technique to avoid learning shortcuts only using the easier to interpret information source, and could again be useful for other tasks involving multimodal inputs.
>
> ### [3] “compares method without controlling for a variety of factors, e.g., LM, LM size, the image encoder, how many tokens are assigned to represent an image”
>
> > We thank the reviewer for this suggestion and now include this in a new table 6 in appendix B, detailing this for all the compared methods. Overall, it can be seen that we are not using a particularly large or powerful image encoder, nor are we using much more image tokens or a bigger LLM. Instead, the main differences are our end-to-end, task specific finetuning via our instruct dataset, combined with our dual branched architecture.

---

> ### Author Response · Authors · 2025-03-11
> **Comments to Final Rating**
>
> We thank the reviewer for responding to our rebuttal. We would like to provide additional results and clarifications to some of the discussion points.
>
> ### [1] “comparison to SoTA LVLMs like Gemini-2, GPT-4o”
>
> > As a quantitative comparison requires manually prompting these models, we initially focused on a qualitative comparison. However, we acknowledge the reviewer’s concern that a single qualitative example does not allow to fully rate the methods. Therefore we updated our comparison and quantitatively evaluated report generation performance for RaDialog, Gemini-2.0 and GPT-4o on 50 randomly sampled x-ray images from our test set. The results are shown in the following table:
> >
> >| Model       | Clinical Efficacy (CE) | BertScore (BS) | BLEU-4 (B-4) | ROUGE_L (R-L) | METEOR |
> |-------------|------------------------|----------------|--------------|---------------|--------|
> | GPT-4o      | 16.3                   | 0.36           | 4.5          | 22.8          | 15.3 |
> | Gemini-2.0  | 27.3                   | 0.35           | 3.3          | 20.0          | 9.7 |
> | RaDialog    | **43.2**               | **0.42**      | **11.9**    | **29.5**      | **16.2** |
> >
> > Similar to our qualitative analysis, these results clearly show that RaDialog performs better by a large margin in all the metrics. If the reviewer finds this comparison valuable, we would happily include it in the print-ready version.
>
> ### [3] “compares method without controlling for a variety of factors, e.g., LM, LM size, the image encoder, how many tokens are assigned to represent an image”
>
> > We would kindly ask the reviewer to again check Table 6, Appendix B in the revised version, where we include and discuss details about the image encoder, LLM size, tokens per image, datasets, as well as frozen and fine-tuned components for all compared models: LLaVA-Med, Rad-FM, XrayGPT, LLM-CXR, CheXagent, R2GenGPT, RaDialog.
> >
> > As a reference this is the table we did include in the appendix:
> >
> >| Method            | Img enc.    | LLM size | Tokens per img | Datasets                     | end-to-end training LLM/img enc. |
> |-------------------|-------------|----------|----------------|------------------------------|----------------------------------|
> | LLaVA-Med         | CLIP-ViT    | 7B       | 576            | PMC-15M                      | ✓ / ×                           |
> | Rad-FM            | 3D ViT      | 13B      | 32             | mix of 18 datasets           | ✓ / ✓                           |
> | XrayGPT           | MedClip     | 7B       | 512            | MIMIC-CXR, IU-Xray           | × / ×                           |
> | LLM-CXR           | VQ-GAN      | 3B       | 256            | MIMIC-CXR                    | ✓ / ×                           |
> | CheXagent         | EVA-CLIP-g  | 7B       | 128            | mix of 28 datasets           | ✓ / ✓                           |
> | R2GenGPT          | Swin-T     | 7B       | 49             | MIMIC-CXR                    | × / ✓                           |
> | RaDialog_align    | BioVil-T    | 7B       | 32             | MIMIC-CXR, RaDialog-Instruct | ✓ / ✓                           |
> | RaDialog_project  | BioVil-T    | 7B       | 196            | MIMIC-CXR, RaDialog-Instruct | ✓ / ✓                           |
>  >
> > If this is not the comparison the reviewer had in mind, we would be grateful for a clarification which additional details the reviewer would like to see about the models.

---

### Official Review · Reviewer_1p4o · 2025-02-12

**Confidence:** 3
**Preliminary Rating:** 4
**Recommendation:** Poster

**Summary:**

This paper presents RaDialog, a vision-language model designed for radiology report generation and interactive assistance. The contributions include a dual-branch architecture focusing on structured clinical findings, a variable instruct training setup to enhance human-AI collaboration and mitigate catastrophic forgetting, and a context-dropping augmentation that emphasizes image information by omitting textual context.

**Strengths:**

I appreciate the writing and easy-to-follow nature. Also, the authors open source the codes and dataset, which are good contributions to the community.

Traditional AI models generate reports in a single pass without interaction. A system that supports dialog-based assistance could be significantly more useful in real-world clinical workflows.

**Weaknesses:**

Is the image encoder frozen? What is the backbone you used?

How does the structured findings extractor improve model interpretability compared to directly training on the structured reports?

Could you highlight the technical contribution? From my understanding, the LVLM is primarily an extension of existing vision-language models (e.g., LLaVA, BLIP-2) with minor adaptations for medical imaging. The structured findings extraction is a straightforward classification task that has been widely explored in prior work on structured radiology reporting. For me, I assume it is combination of existing techniques into this framework to perform more human-AI collaboration.

Experiment:

The decision to fine-tune a 7B Vicuna-based LLM instead of using other models like LLaVA-Med is not justified with efficiency, scalability, or interpretability concerns.

Table 4: It seems like the compared methods are outdated. Please provide more results of studies in 2024.

**Detailed Comments:**

See above.

**Justification Of The Preliminary Rating:**

My major concern is limited novelty. However, the paper still provides valuable contributions to the medical AI community. The public release of the dataset and model promotes reproducibility and allows further advancements in medical vision-language models.

**Questions To Address In The Rebuttal:**

I suggest the authors respond to the weakness I proposed to improve the paper.

**Special Issue:**

No

---

> ### Author Response · Authors · 2025-03-06
> **Addressing the Rebuttal Questions**
>
> We appreciate the reviewer’s recognition of RaDialog’s potential for dialog-based assistance, which could significantly enhance real-world clinical workflows, as well as the value of our public dataset and model release. We address your thoughtful questions below.
>
> ### [1] “Is the image encoder frozen? What is the backbone you used?”
>
> > The image encoder is fine-tuned end-to-end with the rest of the model, building on BioVil-T as the backbone. This can be found in Section 2.1 and Appendix F. We additionally added a new Table 6 in Appendix B listing backbones for all compared methods.
>
> ### [2] “How does the structured findings extractor improve model interpretability?”
>
> > This is an interesting point. By providing the model with high level and accurate findings, we improve both the interpretability and the correctness, as clinicians could interactively review and improve these extracted findings.
>
> ### [3] “Could you highlight the technical contribution?”
>
> > RaDialog stands out from other vision-language models in radiology through its dual-branch architecture, fusing structured findings with image features for improved clinical correctness and introducing context dropping to guide the model to attend to image features. Furthermore, we curate and train with a specialized instruct dataset, mitigating catastrophic forgetting to enable conversational tasks. Unlike single-task LVLMs, this combination delivers SOTA report generation and interactive assistance at the same time.
>
> ### [4] “The decision to fine-tune a 7B Vicuna-based LLM instead of using other models like LLaVA-Med is not justified”
>
> > We thank the reviewer for this interesting question. Indeed, we have internally experimented with LLaVA-Med, but found it to perform no better than LLaVA. Therefore, we opted to use Vicuna as it is a more general model allowing us to retain general knowledge and conversational capabilities of this model.
>
> ### [5] “Table 4: It seems like the compared methods are outdated. Please provide more results of studies in 2024.”
>
> > We’ve updated Table 4 to include recent unpublished 2024 arXiv works, as well as one work we found, which was published after the MIDL submission in 2025. Few 2024 studies on radiology report generation are published in journals or conferences, but we’ve included all relevant ones we found. Considering these newly added comparisons, RaDialog still outperforms in clinical efficacy (CE). We thank the reviewer as these updates clearly improve the quality of our paper.

---

> > ### Comment · Reviewer_eybS · 2025-03-11
> >
> > Thanks for the response. The author has addressed all the issues. I would suggest adding more analysis for the interpretability point (How does the structured findings extractor improve model interpretability compared to directly training on the structured reports?) in the final version. I have increased my point accordingly.

---

### Official Review · Reviewer_eybS · 2025-02-22

**Confidence:** 4
**Preliminary Rating:** 3
**Final Rating:** 4

**Summary:**

The paper introduces RaDialog, a large vision-language model for X-ray reporting and dialog-based assistance. The core innovations are a dual-branch architecture combining visual features and structured findings, and a dialog-based training strategy to prevent catastrophic forgetting. Evaluations show significant improvements over existing methods.

**Strengths:**

1. The technical innovation of combining structured findings extraction with vision-language modeling is well-motivated and effectively implemented. The dual-branch architecture directly addresses a core challenge in medical report generation.
2. The evaluation is comprehensive, spanning multiple datasets (MIMIC-CXR, IU-Xray) and diverse downstream tasks including impression generation, report correction, and Q&A.
3. The inclusion of human evaluation with radiologists (84% preference rate) provides strong validation of the system's clinical utility.
4. The catastrophic forgetting mitigation strategy through instruct dataset design shows careful consideration of practical deployment challenges.
5. The public availability of code and dataset strengthens reproducibility and scientific value.

**Weaknesses:**

1. The paper lacks systematic analysis of failure cases and potential risks. Given the clinical context, understanding when and how the system fails is crucial.
2. The evaluation of computational resources and inference time is limited. For clinical deployment, real-time performance analysis would  be valuable.
3. The technical novelty of this work is limited. Not sure how distinct the work differentiates from other similar Vision-Language models in this domain. The author should clarify this.

**Detailed Comments:**

1. The ablation studies could benefit from more thorough analysis of how each component contributes to final performance.
2. The translation capabilities demonstrated in Figure 3 warrant more detailed evaluation and discussion.
3. The context dropping augmentation technique deserves more theoretical justification.
4. The radiologist evaluation protocol and criteria should be described in more detail.

**Justification Of The Final Rating:**

The author has clarified the weakness addressed in both rebuttal and the revisions. The revised version has systematically increased the quality of the manuscript. Therefore I think its fair to increase my score from 3 to 4.

**Justification Of The Preliminary Rating:**

The paper presents significant technical innovations with clear clinical relevance and strong empirical validation. The dual-branch architecture and dialog-based training strategy effectively address key challenges in medical report generation. The thorough evaluation across multiple tasks and datasets, combined with public code release, makes this a valuable contribution to the field. However, the work lacks technical novelty and deep analysis on the performance and their approach efficiency.

**Questions To Address In The Rebuttal:**

1. How does the system handle cases where the structured findings extractor and visual features provide conflicting information?
2. What safeguards ensure patient privacy when retrieving similar cases?
3. How generalizable is the approach to other types of medical imaging beyond chest X-rays?
4. How distinct is this work compared with other medical vision-language models in Chest X rays?

---

> ### Author Response · Authors · 2025-03-06
> **Addressing the Rebuttal Questions Set1**
>
> We thank the reviewer for recognizing the innovation of our dual-branch architecture, the comprehensiveness of our evaluation, the utility of our instruct dataset in mitigating catastrophic forgetting, and the value of our open-source release. We address your insightful comments below.
>
> ### [1] “paper lacks systematic analysis of failure cases and potential risks”
>
> > We appreciate this valuable feedback. In the updated paper, we added a new analysis of the limitations of the method at the end of the Section 4.: We performed an additional failure case analysis on 50 RaDialog predictions on the MIMIC-CXR test set. We observe that in general, Radialog performs well in identifying major radiological findings and the presence of support devices, showcasing its ability to detect significant abnormalities and medical hardware. The most frequent errors are mischaracterizing less frequent findings, misjudging the severity of conditions, and occasionally introducing minor incorrect details such as the exact position of findings or devices. While not flawless, Radialog can provide valuable utility for handling simpler cases, providing initial drafts and enabling collaborative radiologist-AI reporting.
>
> ### [2] “evaluation of computational resources and inference time is limited”
>
> > Thank you for pointing this out. We’ve added inference times (tokens/sec and sec/report) for both models in Section 4, and updated Appendix E with GPU details.
>
> ### [3] “technical novelty of this work is limited.”
>
> > RaDialog stands out from other vision-language models in radiology through its dual-branch architecture, fusing structured findings with image features for improved clinical correctness and introducing context dropping to guide the model to attend to image features. Furthermore, we curate and train with a specialized instruct dataset, mitigating catastrophic forgetting to enable conversational tasks. Unlike single-task LVLMs, this combination delivers SOTA report generation and interactive assistance at the same time.
>
> ### [4] “The ablation studies could benefit from more thorough analysis of how each component contributes to final performance”
>
> > Thank you for this suggestion. We agree this ablation is crucial and therefore moved our ablation study from Appendix D.3 to Section 4 (Table 2), showing how each component boosts performance. The instruct dataset’s impact is evident in Tables 3 and 4, with the instruct variant clearly outperforming the non-instruct one in interactive downstream tasks, while maintaining report generation performance (Table 1).
>
> ### [5] “context dropping augmentation technique deserves more theoretical justification”
>
> > We thank the reviewer for this comment and have expanded Section 2.2 with a more detailed justification.: Context Dropping simulates inaccuracies in the initial report by varying textual input completeness. This prevents over-reliance on text, which is easier for the model to interpret. Training with different levels of textual availability strengthens visual reasoning by promoting feature extraction from both visual and textual inputs.
>
> ### [6] “The radiologist evaluation protocol and criteria should be described in more detail.”
>
> > We fully agree and have now added these details in Section 4, “Qualitative Conversation Results.”: For each image, both models generated a report and performed one of several conversational follow-up tasks. A board-certified radiologist, blinded to report sources, reviewed the original X-ray and the two reports in randomized order. They completed a structured questionnaire per case, assessing (1) report correctness, (2) conversational performance, and (3) overall preference.

---

> ### Author Response · Authors · 2025-03-06
> **Addressing the Rebuttal Questions Set2**
>
> ### [7] “How does the system handle cases where the structured findings extractor and visual features provide conflicting information?”
>
> > RaDialog is trained with predicted structured findings, allowing it to learn how to handle inaccuracies in the structured finding extractor. Our ablation study (now added in Table 2) confirms that both components improve performance, demonstrating they work effectively together. Additionally, we provide the following two qualitative examples showing that RaDialog can override structured findings when necessary:
> >
> > Example 1:
> >
> >Predicted Structured Findings: Lung opacity
> >
> >Generated Report: Left-sided Port-A-Cath tip terminates in the mid SVC. [...] Increased interstitial markings are noted diffusely, most pronounced within the lower lobes. [...]
> >
> >→ RaDialog correctly identifies the catheter, even though external devices are not mentioned in the structured findings.
> >
> > Example 2:
> >
> >Predicted Structured Findings: Cardiomegaly, fracture
> >
> >Generated Report: The cardiac and mediastinal silhouettes are stable with the cardiac silhouette enlarged. No focal consolidation is seen. There is no pleural effusion or pneumothorax.
> >
> >→ RaDialog correctly ignores the "Fracture" finding, demonstrating its ability to override structured findings when necessary.
>
> ### [8] “What safeguards ensure patient privacy?”
>
> > Our approach ensures patient privacy by operating entirely within the hospital infrastructure. Unlike cloud-based models such as ChatGPT, RaDialog does not require external data transmission. This allows sensitive patient information to remain secure and compliant with hospital data policies.
>
> ### [9] “How generalizable is the approach to other types of medical imaging beyond chest X-rays?”
>
> > Our model is easily adaptable to other modalities, as our publicly available code allows for straightforward re-training, primarily limited by data availability. The structured findings branch can leverage CheXbert if radiology reports exist or directly use structured labels if available. Additionally, our instruct dataset creation process and prompt design, detailed in the paper, provide a clear framework for generating similar datasets for other modalities. Overall, we believe the core concepts of our approach are widely applicable and transferable.

---

### Author Rebuttal · Authors · 2025-03-06

**Rebuttal:**

We sincerely thank the reviewers for their insightful and constructive feedback. We appreciate the recognition of our dual-branch architecture, instruct dataset, and extensive evaluation, as well as the potential impact of RaDialog for real-world clinical applications. The reviewers’ comments have helped us improve the clarity, completeness, and rigor of our work.

We upload a revised manuscript with the following changes:

- Section 2.2 \- “Context Dropping Augmentation”: improved justification for context dropping
- Section 4 \- “Radiology report generation”: inference time measurements
- Section 4 and Table 2: added section for “Ablation of Architectural Components”
- Section 4 \- “Qualitative Conversation Results”: added detailed radiologist evaluation protocol
- Section 4: added “Limitations” section, discussing possible failure cases
- Appendix A: Extended Table 1 with more recent report generation works
- Appendix B / Table 6: Added details on LVLM configurations
- Appendix D / Figure 4: updated qualitative comparison to include GPT-4o and Gemini-2.0
- Appendix F: added GPU memory details

All changes in the revision are marked in light blue.

We also provide detailed comments for all three reviewers regarding their concrete questions.

**Supporting Material:**

/attachment/ae0871123e05ed240568e105f5f60dfb4586e1af.pdf

---

### Meta-Review · Area_Chair_NJT2 · 2025-03-22

**Recommendation:** Accept (Poster)
**Confidence:** 5

**Metareview:**

This paper presents RaDialog, a novel large vision-language model (LVLM) designed for radiology report generation and interactive dialog-based assistance. The core contributions include: (1) a dual-branch architecture that integrates visual features from X-ray images with structured clinical findings, enhancing the model’s ability to synthesize detailed reports; (2) a variable instruction-tuning strategy that balances human-AI collaboration while mitigating catastrophic forgetting during training; and (3) a context-dropping augmentation technique, which encourages the model to prioritize image-derived information by selectively omitting textual context. The authors adapt and refine existing frameworks (e.g., LLaVA and BLIP) using a curated instruction dataset, demonstrating significant improvements over state-of-the-art methods across diverse tasks and evaluation settings.

All reviewers agreed that the authors addressed their initial concerns during the rebuttal phase, and they all recommended acceptance.